# Plant Growth Promotion Function of *Bacillus* sp. Strains Isolated from Salt-Pan Rhizosphere and Their Biocontrol Potential against *Macrophomina phaseolina*

**DOI:** 10.3390/ijms22073324

**Published:** 2021-03-24

**Authors:** Stefany Castaldi, Claudia Petrillo, Giuliana Donadio, Fabrizio Dal Piaz, Alessio Cimmino, Marco Masi, Antonio Evidente, Rachele Isticato

**Affiliations:** 1Department of Biology, University of Naples Federico II, Complesso Universitario Monte S. Angelo, Via Cinthia 4, 80126 Naples, Italy; stefany.castaldi@unina.it (S.C.); claudia.petrillo@unina.it (C.P.); 2Department of Pharmacy, University of Salerno, 84084 Fisciano, Italy; gdonadio@unisa.it; 3Department of Medicine, Surgery and Dentistry, University of Salerno, Via Giovanni Paolo II, 84084 Fisciano, Italy; fdalpiaz@unisa.it; 4Department of Chemical Sciences, University of Naples Federico II, Complesso Universitario Monte S. Angelo, Via Cinthia 4, 80126 Naples, Italy; alessio.cimmino@unina.it (A.C.); marco.masi@unina.it (M.M.); evidente@unina.it (A.E.)

**Keywords:** plant growth-promoting bacteria, spore-forming bacteria, *Bacillus vallismortis*, *Macrophomina phaseolina*, phenotypic and genotypic characterization, biocontrol agents

## Abstract

In recent decades, intensive crop management has involved excessive use of pesticides or fertilizers, compromising environmental integrity and public health. Accordingly, there has been worldwide pressure to find an eco-friendly and safe strategy to ensure agricultural productivity. Among alternative approaches, Plant Growth-Promoting (PGP) rhizobacteria are receiving increasing attention as suitable biocontrol agents against agricultural pests. In the present study, 22 spore-forming bacteria were selected among a salt-pan rhizobacteria collection for their PGP traits and their antagonistic activity against the plant pathogen fungus *Macrophomina phaseolina*. Based on the higher antifungal activity, strain RHFS10, identified as *Bacillus vallismortis*, was further examined and cell-free supernatant assays, column purification, and tandem mass spectrometry were employed to purify and preliminarily identify the antifungal metabolites. Interestingly, the minimum inhibitory concentration assessed for the fractions active against *M. phaseolina* was 10 times lower and more stable than the one estimated for the commercial fungicide pentachloronitrobenzene. These results suggest the use of *B. vallismortis* strain RHFS10 as a potential plant growth-promoting rhizobacteria as an alternative to chemical pesticides to efficiently control the phytopathogenic fungus *M. phaseolina*.

## 1. Introduction

In the last century, the world population reached a size three times greater than any previous value across the whole history of humanity. To cope with the rising request for nutrients, such as those provided by wheat and rice, current agricultural practices are based on the wide use of chemical fertilizer and pesticides. As a result, agrochemical multinationals have gradually acquired the control of global food production and modern agriculture is increasingly diverging from the traditional model [1]. Additionally, the extensive use of synthetic agrochemicals has generated heavy environmental pollution and serious risk for human and animal health due to their translocation along the food chain [1,2]. The massive use of pesticides has also led to a gradual loss of protection efficiency due to new resistances acquired by pests, with a continuous increase in pesticide dosage [2,3]. A sustainable and safe strategy to ensure crop production is to substitute agrochemicals with Plant Growth-Promoting Rhizobacteria (PGPR) as agents stimulating plant growth and health [3,4,5]. These beneficial microbes not only play an important role in increasing soil fertility but also enhance the growth and vigor of the plants—PGPRs, by colonizing the roots, may enhance nutrient uptake by nitrogen fixation or P solubilization [4], reduce abiotic stresses by biofilm production [5] or regulate plant hormone production [4]. Emerging evidence has shown that rich microflora of the rhizosphere can reduce plant disease through several antagonistic mechanisms such as competition, the production of cell-wall-degrading enzymes, (e.g., chitinase, glucanase, and protease) [6], volatile compounds and siderophores [7], antibiosis or the induction of plants’ systemic resistance [8]. Replacing agrochemicals with the application of PGPRs may have both economic and environmental impacts, including relevant benefits such as rising yields, reduction in or elimination of chemical residues, limited or no development of resistance by pests and pathogens, employment of agricultural raw materials, and a low risk to nontarget organisms, including pollinators. For this reason, intensive research on this group of microorganisms has been taking over to develop new biofertilizers and biocontrol agents.

In this contest, *Bacillus* genera include several exo- and endophytic bacteria species and plant growth-promoting (PGP) features have been associated with different strains [9,10]. In addition to the benefits shared with other PGPR, such as solubilization of soil P, enhancement of nitrogen fixation, and siderophore production, *Bacillus* spp. are suitable as biofertilizers because: (i) their application has little, if any, effect on the composition of the soil microbial communities, being common members of the plant root microflora [11]; (ii) these bacteria may form endospores, which can survive at high temperatures and dehydration, making the formulation of a commercial product easier [12]; (iii) some *Bacillus* PGPR strains have also been reported to perform well under different environmental conditions [13]. As biocontrol agents, *Bacillus* spp. exhibit both direct and indirect mechanisms to suppress diseases caused by pathogens. These bacteria secrete a vast range of secondary metabolites, such as cell-wall-degrading enzymes, and antioxidants that assist directly the plant in its defense against pathogen attack [14]. As an indirect mechanism, *Bacillus* spp. are able to induce the acquired systemic resistance of the colonized plant [8].

This manuscript describes the screening of 22 *Bacillus* strains isolated from samples of the rhizosphere of *Juniperus sabina* [15] collected from the National Park of Ses Salines d’Eivissa, Formentera (Spain), focused on finding a PGPR strain with antagonistic activity against the phytopathogenic fungus *Macrophomina phaseolina*. 

*M. phaseolina* (Tassi) Goid is responsible for charcoal root rot, the most common and widely spread root disease affecting more than 500 cultivated and wild plant species. The fungus is distributed worldwide and prevalently in arid areas with low rainfall and high temperature where it can survive for up to 15 years in the soil as a saprophyte [16]. *M. phaseolina* generally affects the fibrovascular system of the roots and basal internodes producing black sclerotia, which allow the fungus survival after the plants rotted [16].

Each year, this fungus induces heavy damages in agrarian plants with a high world market value, such as soy, sunflower, leguminous, and corn [16]. Soybean grains, in particular, are globally utilized not only as foods but also as substrates for feeds, fuels, and bio-based materials [17]. Thus, many efforts are made for the control of *M. phaseolina* to reduce or avoid the loss of agricultural yields and the consequent economic damage.

Additionally, PGPRs have been evaluated as biocontrol agents against *M. phaseolina* and strains belonging *to Pseudomonas* and *Bacillus* genera showed the best performance. In a study carried out by Simonetti et al. [18], two strains, namely *Pseudomonas fluorescens 9* and *Bacillus subtilis 54*, have been assayed for antifungal activity in combination with manganese phosphite or alone and shown to significantly reduced soybean disease severity induced by *M. phaseolina* compared to the untreated control.

Several studies are still in progress to identify the main antifungal metabolites produced by PGPRs and clarify their modes of action to achieve optimum disease control.

## 2. Results

### 2.1. Isolation and Screening of Plant Growth-Promoting Spore-Forming Rhizobacteria

Aerobic spore-forming bacteria were isolated from rhizosphere samples of *J. sabina* collected in Parque Natural de Ses Salines d’Eivissa, Formentera (Spain), as described in the Materials and Methods section. A preliminary characterization based on the bacterial morphology and growth properties has allowed the selection of 22 facultative anaerobic strains, mesophiles, which are able to grow at a different pH range (Appendix A).

Analysis of the DNA sequence of the 16S RNA gene of the 22 strains allowed the identification of all of them as belonging to the *Bacillus* genus (Appendix A). In order to confirm the different species obtained by BlastN analysis (Appendix A), a phylogenetic analysis (Figure 1) was performed by comparing the 16S sequences with respective type strains (^T^) available at the NCBI Taxonomy database. The analysis corroborated the different *Bacillus* species by >0.90 bootstrap values. All isolates belong to species commonly considered as PGPR for their ability to colonize roots [11,19] and produce antimicrobial compounds [14,19].

The selected strains were analyzed for PGP traits by testing the presence of both fertilizing and biocontrol features. As summarized in Table 1, a high proportion was able to solubilize phosphate (Appendix A), produce siderophores (Appendix A) and indoleacetic acid, while only some of the strains were biosurfactant and biofilm producers and showed swarming motility.

Then, the potentiality as biocontrol agents of the 22 strains was tested analyzing their ability to secrete lytic enzymes (Appendix A) [20]. As shown in Table 1, the number of protease and xylanase producers was the highest (over 90%) followed by amylase, chitinase and cellulase producers (over 80%), whereas less than 50% were lipase- producers (45%).

### 2.2. Antagonistic Activity of Spore-Forming Isolates against Fungal Plant Pathogen

The antagonistic activity of the 22 strains was examined against the phytopathogen *M. phaseolina* by dual-culture assay (Figure 2A).

Based on the size of the inhibition zone in dual-culture tests, some strains were found to be highly efficient against the fungal pathogen while others had limited or no antimicrobial activity (Figure 2B). For a more detailed analysis, the produced inhibition halos were observed under a stereomicroscope, highlighting agar-diffusible antifungal molecule production by the most active strains (Figure 2A, panel 3; Appendix A).

Of all analyzed isolates, RHFS10 and RHFS18 proved to higher potentiality than PGPR, since they possess traits beneficial for both plant growth, such as the ability to solubilize phosphorus or produce siderophores, and show antagonistic ability against phytopathogens. For these reasons, both strains were selected for further experiments. Strain RHFS28, able to produce lytic enzymes but not showing antifungal activity, was selected as a negative control for the next experiments.

To assess the effect of the cell-free culture supernatants (CFSs) of RHFS10 and RHFS18 on mycelial growth, the CFSs at 24, 48, 72 and 96 h were collected and tested against *M. phaseolina*. The commercial fungicide pentachloronitrobenzene (PCNB) dissolved in toluene was used as a positive control and toluene alone was used as a negative control of the experiments (Figure 3A). The antifungal activity increased proportionally with the growth time reaching a maximum after 72 h, specifically for the RHFS18 strain (Figure 3B). Based on the efficiency of inhibition, measured by the percentage of mycelial growth reduction, strain RHFS10 was chosen for further investigation.

### 2.3. Characterization of Antifungal Metabolites

The stability of the antifungal metabolites secreted by RHFS10 was tested by incubating the CFS collected after 72 h (72-CFSs) with different proteolytic enzymes or organic solvents and then tested for inhibition of mycelial growth.

As shown in Figure 4A, the 72-CFS still had notable activity after incubation with organic solvents but decreased under the action of proteinase K or pepsin.

Thermostability was verified incubating the 72-CFS at increasing temperatures for 1 or 3 h. The results showed that treatments at 65 and 75 °C do not affect the inhibitory effect against *M. phaseolina*, while at 85 °C a reduction in the antifungal activity was observed (Figure 4B).

Finally, metabolites of the 72-CFSs were extracted with ethyl acetate at pH 2.0 and pH 7.0 and the two obtained phases were separated and tested against *M. phaseolina*. The results showed that the antifungal activity was mainly associated with the aqueous phase at pH 7.0 (data not shown). This data indicated a protein nature of the bioactive molecules in agreement with the protease sensitivity recorded in the previous tests.

### 2.4. Purification of Antifungal Metabolites

To preliminarily identify the antifungal compounds released by the RHFS10 strain, 72-CFS was subjected to purification by two different steps. First, the 72-CFS was fractionated and the obtained fractions were tested against *M. phaseolina*. As shown in Figure 5A, the antifungal activity was observed in the fraction containing compounds with molecular weights between 10 and 50 kDa. In the second step of purification, the polypeptides present in 72-CFS were collected with ammonium sulfate, dialyzed to eliminate the polypeptides with a molecular weight lower than 10 kDa, and subjected to column chromatography. The three obtained fractions were tested against *M. phaseolina* and peaks 1 and 2 showed a wide zone of inhibition while no antagonistic activity was detected for the metabolites recovered in peak 3 (Figure 5B).

#### 2.4.1. Minimum Inhibitory Concentration 

Minimum inhibitory concentration (MIC) of the antifungal compounds presents in peaks 1 and 2 was determined, incubating decreasing concentrations of peaks 1 and 2 (Figure 6(A1,A2)) with *M. phaseolina* plugs. The antifungal efficiency of the compounds present in the peaks was compared to the commercial fungicide PCNB (Figure 6(A4)). The results obtained after 5 days of incubation clearly showed higher antifungal activity of peaks 1 and 2 than the fungicide PCNB. In particular, the deduced MIC for both peaks was 50 µg/mL, 10 times less than that deduced for PCNB (0.5 mg/mL). We also compared the stability of the antifungal activity over time. In this regard, the bioactive compounds present in peaks 1 and 2 perfectly retained their fungal growth inhibition for up to 14 days, while PNCB’s efficiency decreased after a week. Peak 3 confirmed its inactivity (Figure 6 (A3)).

#### 2.4.2. Preliminary Identification of Bioactive Compounds

Finally, the three fractions were analyzed by liquid chromatography coupled with tandem mass spectrometry (LC–MS/MS). As shown in Table 2, several protease and lytic enzymes were identified in the two antifungal active peaks. Two different forms of subtilisin-like proteins were identified in peak 1, showing apparent molecular weights of 39 and 28 kDa and corresponding to the mature serine-protease and the proenzyme, respectively. Additionally, the glucuronoxylanase XynC was also detected. Both subtilisin-like protein forms were also present in peak 2, even if with a lower concentration, together with a B-glucanase, whereas peak 3 contains a metalloprotease and an alpha-amylase. As serine-proteases, beta-glucanase and glucoronoxylanase were demonstrated to act as antifungal agents [21,22], our results suggest that the activity of these secreted metabolites could be responsible, at least partially, for the antifungal action of RHFS10. To further corroborate this hypothesis, a mass spectrometry-based proteomic analysis on the previously described 72-CFSs of RHFS10 strain treated at increasing temperatures (cfr. 3.4) was performed. Again, the two forms of subtilisin and glucuronoxylanase XynC were identified in the samples retaining the antifungal activity. Interestingly, the two proteins were not detected in CFS from the negative control (RHFS28) when subjected to the same treatment. Although the genome of RHFS10 was in permanent draft stage (SAMN17389611), it allowed us to confirm the presence of all the purified protein genes, which when expressed could be involved in inhibiting fungal growth.

## 3. Discussion

Fungal pathogens represent one of the most common causes of plant disease and are responsible for losing a third of crops annually [23], causing economic loss and impacting global poverty. Among phytopathogenic fungi, *M. phaseolina* (Tassi) Goid is one of the most virulent and dangerous plant pathogens. The fungus is responsible for charcoal rot disease and for the consequent significant yield losses in major crops such as maize, sorghum, soybean, and common beans each year. The harmfulness of the pathogen is due to its ability to produce phytotoxins, to survive for a long time in the soil, and to target any stage of plant growth affecting seeds, seedlings, and adult plants [24]. The persistence of *M. phaseolina* in the soil and in turn its capacity to trigger plant infection depends on its ability to compete with other microorganisms of the rhizosphere—for example, competing for organic sources or host root colonization. For this reason, a growing number of studies have been focusing on the isolation and characterization of PGPRs able to limit *M. phaseolina* growth. PGPRs can not only colonize the rhizosphere improving plant growth by enhancing nutrient uptake or regulating plant hormone production, but can suppress a broad spectrum of phytopathogens, producing different antagonistic compounds or competing for nutrients.

In this contest, the focus of our research was to identify promising Bacilli rhizobacteria acting as biofertilizers and biocontrol agents against *M. phaseolina*. *Bacillus* species are a major type of rhizobacteria able to be beneficial to plants and to perform the same role as chemical fertilizers [25] and pesticides [26]. As PGPR, *Bacillus* spp. act both by direct and indirect mechanisms, secreting phytohormones, antioxidants, solubilizing soil P, enhancing nitrogen fixation, or producing cell-wall-degrading enzymes and siderophores that promote plant growth and suppress the pathogens [27].

Moreover, the ability of the *Bacillus* spp. the produce endospores makes them more suitable candidates for PGPR-based commercial products since the resistance features of the spores can ensure the persistence of the bacteria during industrial processing and after their spread in the environment [12].

To this aim, spore-forming bacteria were isolated from salt-pan rhizosphere (Formentera, Spain) of the nurse plant *J. sabina*. As a nurse plant, *J. sabina* ensures a beneficial organization of plant communities and maintenance of biodiversity, particularly in harsh environments [28]. Growing evidence highlights that nurse plants alter the composition of soil bacterial communities, selecting microbiota that are more effective at nutrient mineralization and involved in plant growth-promoting mechanisms. Among isolates, 22 spore-forming bacteria strains were identified at a species level and first screened for their plant growth-promoting traits. More than 50% of the selected strains have shown to solubilize insoluble phosphates, to produce siderophores and secrete IAA, the main plant auxin able to regulate growth and developmental processes. These findings confirm that the rhizosphere of nurse plants is a useful source of PGPRs. Then, the biocontrol activity against the fungus *M. phaseolina* has been tested by dual-culture assay.

Among the 22 isolates, strain RHFS10, identified as *B. vallismortis*, showed the best performance for plant growth-promoting applications both as biofertilizer and biocontrol agents. The fungal growth inhibition revealed in the cell-free supernatant assay suggested the secretion of antifungal extracellular metabolites not induced by direct contact with the fungus. These data were in agreement with the stereoscopic observation of coculture experiments. Additionally, the antagonist activity of RHFS10 was not influenced by the bacterial growth stage, suggesting a constitutive production of the antimicrobial compounds.

Stability experiments revealed a thermostability of the antifungal compounds up to 75 °C and resistance to various organic solvents. Instead, the sensitivity to protease treatment as well as the association of the antifungal activity with the aqueous phase during the extraction with an organic solvent suggests a proteinaceous nature of the metabolites.

Purification experiments have associated the antifungal activity with metabolites with molecular weights between 10 and 50 kDa, while LC–MS/MS analysis revealed the presence of proteases and hydrolytic enzymes in the active fractions. In particular a glucuronoxylanase of 45 kDa and a homologous of the serine protease Subtilisin NAT from *B. subtilis subsp. natto* that could be directly implicated in the fungal growth inhibition. Both proteins were absent in the inactive peak, confirming their involvement in the observed antifungal activity.

There are, indeed, several functions ascribed to the release of these compounds during the stationary phase of growth. It is well known that during this very phase of their life cycle, bacteria generally release hydrolytic enzymes mainly involved in the cell wall turnover and nutritional functions, which in many cases show antimicrobial and/or antibiofilm activity [29]. Moreover, it has been lately reported that subtilisin-like proteases and glucuronoxylanases can digest fungal cell wall structural proteins [30], supporting our preliminary results. Recently, it has been shown that *B. subtilis natto* can use several fungal materials as a carbon source for growth, pointing out the role of constitutively secreted protease as a nutrient scavenger as well as a potent tool for fungal biocontrol [31].

A further important result is the higher efficiency of the purified antifungal metabolites than the commercial fungicide PCNB, used as a positive control in antagonism assays. The minimum inhibitory concentration assessed for the bacterial bioactive compounds against *M. phaseolina* growth (50 µg/mL) was 10 times lower than the one estimated for the commercial fungicide PCNB (0.5 mg/mL). Interestingly, the bacterial metabolites also appeared to be more stable over time—they retained their antifungal activity for up to two weeks, while PCNB registered an efficiency reduction after 6 days only. Hence, the purified bacterial bioactive metabolites might be employed in lower concentrations, reaching a higher long term efficiency compared to chemical fungicides.

Altogether, these results suggest a strong antifungal effect of the protein compounds produced by the RHFS10 strain and a promising prospect for agricultural applications. The bacterial bioactive proteins could represent a valid sustainable eco-friendly fungicide and have potential as a biocontrol agent as an alternative to chemical pesticides.

Future studies will focus on the effect of the *M. phaseolina* on the expression of antifungal metabolites produced by RHFS10, to verify if the fungus itself may enhance the production of the bioactive compounds already detected in this study or, perhaps, trigger the expression of new metabolites. Other studies also need to optimize their large scale production and to find their best formulation for their application in field.

## 4. Materials and Methods

### 4.1. Isolation of Bacteria

Samples of the rhizosphere of *Juniperus sabina* plants were collected from the National Park of Ses Salines d’Eivissa, Formentera (Spain). To isolate rhizospheric bacteria, 1 g of roots samples was washed three times with 2 mL sterile distilled water to remove impurities, transferred into 9 mL 1× PBS, and vortexed. The selection of spore-forming strains was promoted through a heat pretreatment at 80 °C to kill all vegetative cells. In total, 1 mL of the mixture was inoculated into 9 mL of LB (8 g/L NaCl, 10 g/L tryptone, 5 g/L yeast extract), serially diluted up to 10–6 and 0.1 mL of each dilution were spread on LB agar plates. Plates were incubated at 30 ± 1 °C for 2–3 days. Pure cultures were obtained by serial subculturing. Glycerol stocks of the isolates were prepared and stored at −80 °C.

### 4.2. Growth Conditions

Each bacterial isolate was characterized by visual inspection for colony color and morphology, such as colony shape, size, margin and appearance. The ability to grow in facultative anaerobic conditions was determined using the AnaeroGen sachets (Unipath Inc., Nepean, ON, Canada) placed in a sealed jar with bacteria streaked on LB agar plates and incubated at 37 °C for 3–4 days. To determine the optimum growth conditions, the bacterial isolates were grown in LB agar at different pH (2.0, 4.0, 6.0, 7.0, 8.0, 10.0, 12.0) [32] and temperature (4, 15, 25, 37, 50, 60 °C) ranges [33]. Plates were incubated until the appearance of bacterial colonies.

### 4.3. Isolates Identification by PCR Amplification of 16S rRNA

Exponentially growing cells were used to extract chromosomal DNA using the DNeasy PowerSoil kit (Qiagen, Hilden, Germany) according to the manufacturer’s instructions. 16S rRNA gene was PCR amplified by using chromosomal DNA as a template and oligonucleotides forward 8F (5′-AGTTTGATCCTGGCTCAG-3′ annealing at position + 8⁄+ 28) and reverse 1517R (5′-ACGGCTACCTTGTTACGACT-3′ annealing at position + 1497⁄+ 1517). These two oligonucleotides were designed to amplify a 1500 bp DNA fragment and the reaction was carried out according to Grönemeyer et al. [34] in an Esco SwiftTM MaxPro Thermal Cycler. The 1500 bp DNA amplified fragment was sequenced at the Bio-Fab research sequencing facility and analyzed using Basic Local Alignment Search Tool (BLAST). Phylogenetic analyses were carried out using Seaview 4.4.0 software package (http://pbil.univ-lyon1.fr/software/seaview.html, accessed on 7 January 2020) on 16S ribosomal RNA genes aligned using the Muscle algorithm. All 16S rRNA sequences were deposited in the NCBI Sequence Read Archive and identified with the accession number as shown in Appendix A.

Phylogenetic reconstruction for nucleotide alignment was carried out using the maximum likelihood algorithm (PhyLM). The gene sequences of the isolated bacteria were aligned to the representative type strains (^T^) belonging to the same species obtained from BlastN analysis. The percentage of replicate trees in which the associated taxa clustered together in the bootstrap test (1000 replicates) is shown next to the branches.

### 4.4. In Vitro Screening for Plant Growth-Promoting (PGP) Traits

#### 4.4.1. Phosphate Solubilization

The ability to solubilize inorganic phosphate was tested by growing the bacterial isolates on Pikovskaya agar (Oxoid Ltd., Hampshire, UK) dyed with bromophenol blue [35] for 10 days at 30 °C. The formation of more transparent zones around the bacterial colonies was indicative of inorganic phosphate solubilization on Pikovskaya agar.

#### 4.4.2. Siderophore Production

To test siderophores production, 3 µL of overnight-grown culture in LB medium was spot-inoculated on iron-free S7 agar minimal medium. After 72 h of incubation at 28 °C, 10 mL of Chrome Azurol S (CAS) agar medium [36] was applied over agar plates containing cultivated microorganisms. Development of yellow-orange halo zone around bacterial spots was observed after 1 h of incubation.

#### 4.4.3. Indole Acetic Acid Detection

To detect the IAA production, the bacteria were grown in LB broth for 72 h a 37 °C with shaking at 150 rpm. After, 2 mL of bacteria supernatant was mixed with 4 mL of Salkowski reagent (0.5 M FeCl_3_ in 35% HClO_4_ solution) and 2 drops of orthophosphoric acid, and was finally incubated for 30 min at 25 °C. The development of pink color indicates IAA production [37].

#### 4.4.4. Biosurfactant Production

The bacterial strains were spot-inoculated on blood agar plates (BBL™ Trypticase™ Soy Agar (TSA II) with 5% Horse Blood) and after 72 h of incubation at 28 °C, the clear zone around the colonies indicates a positive result [38].

#### 4.4.5. Swarming Motility

Bacterial isolates were analyzed for their swarming motility using LB with spot-inoculation on agar 0.7% and incubated at 37 °C overnight.

#### 4.4.6. Biofilm Production

To evaluate the ability to produce biofilm, the isolates were separately grown in glass tubes in LB medium as described by Haney et al. (2018) [39]. Cultures were inoculated by adding 10 µL of an overnight culture of bacteria into 1 mL of sterile media, and the tubes were incubated statically at either 37 °C for 48 h.

### 4.5. Evaluation of Potential Biocontrol Features

#### 4.5.1. Screening for Hydrolytic Enzymatic Activity

Twenty-two bacterial isolates were grown separately in 5 mL of LB broth a 37 °C overnight with shaking at 150 rpm. In total, 3 µL of each fresh bacterial culture was spot-inoculated on different assay plates to test hydrolytic enzyme activity. The protease activity was performed on Skimmed Milk Agar (SMA) [40] and the lipase activity on Tributyrene Agar medium [41]. After overnight at 37 °C, the formation of a clear halo around the colony was considered as positive production of these enzymes. To detect the amylase activity was used the method described by Sethi et al. (2013) [42] with Starch Agar plates. After the overnight incubation at 37 °C, the plates were flooded with iodine solution and the hydrolysis of starch was observed as a colorless zone with a violet background around grown colonies. For the detection of cellulase and xylanase activities, Xylanase Production Medium (XPM) agar plates were used with 0.5% xylan [43] (Megazyme) and a minimal medium with 0.5% carboxymethylcellulose (CMC) [44] as a sole carbon source. The plates were incubated at 37 °C for 3 days after which hydrolysis zones were visualized by flooding the plates with 0.1% Congo Red for 15–20 min and then destained by washing twice with 1 M NaCl. Plates, where CMC and xylan were omitted, were used as nonsubstrate controls. Transparent hydrolytic zones around the colonies were considered positive. For the chitinase activity, the bacterial strains were spot-inoculated on colloidal chitin-containing medium plates [45]. After incubation at 25 ± 2 °C for 2–3 days, the clear zones around or within the colonies are considered positive evidence. The catalase activity was checked qualitatively as described by Geetha et al. (2014) [46]. Three percent H_2_O_2_ was added (3–4 drops) on the colonies grown on LB agar plates; effervescences of O_2_ released from the bacterial colonies indicate the positivity of catalase activity.

All experiments were performed in triplicate.

#### 4.5.2. Dual-Culture Assay

The isolated strains were examined in vitro for antifungal activity against pathogenic fungus *M. phaseolina* (Tassi) Goid (ATCC^®^ 64334™). The fungus was obtained from infected soybean roots growing in Pergamino, Buenos Aires, Argentina, and it was maintained on Potato Dextrose Agar (PDA) in Petri dishes.

The in vitro antifungal bioassays were carried out based on the dual-culture method as previously described by Khamn et al. (2009) [47] with some modifications.

Fungal plugs of 6 × 6 mm diameter were placed at the center of PDA plates and 5 µL of bacteria strains overnight grown in LB broth was placed on the opposite four sides of the plates at 1.5 cm away from the fungal disc. Plates containing the fungal plugs without bacterial inoculation were used as control plates. All plates were incubated at 28 °C for five days. The percentage of inhibition of the fungal growth was calculated using the following formula:% = [(Rc − Ri)/Rc] × 100
where Rc is the radial growth of the test pathogen in the control plates (mm), and Ri is the radial growth of the test pathogen in the test plates (mm). The experiment was repeated thrice. Bacterial strains that showed an inhibition of the growth of pathogenic fungus were observed by stereoscopic microscope 10× magnification.

#### 4.5.3. Antifungal Assay of Cell-Free Supernatants (CFSs)

Bacteria were grown on LB at 28 ± 2 °C and aliquots of the suspensions, collected at 24 h intervals for the first 96 h. Cells were removed by centrifugation (7000× *g* for 30 min) and supernatants were filtered using 0.22 μm-pore-diameter membranes (Corning^®^) and concentrated 1:10. Then, 20 μL aliquots of sterilized supernatant samples were placed on the opposite four sides of the PDA plate at 1.5 cm from the fungal disc (6 × 6 mm diameter) of *M. phaseolina* [48]. As a positive control, fungicidal pentachloronitrobenzene ≥ 94% (PCNB) (Sigma-Aldrich, Saint-Louis, MO, USA) dissolved in toluene was used. Toluene alone was used as a negative control. Plates were prepared in triplicate, incubated at 28 °C for 5 days, and examined for zones of inhibition of grown colonies.

### 4.6. Extraction of Secondary Metabolites

The strains were grown in 300 mL of LB at 28 ± 2 °C and for 72 h. The broth cultures were then centrifuged at 9000× *g* for 30 min at 4 °C and filtered through a 0.22 µm syringe filter. The culture filtrate was extracted at pH7 and pH2 three times for each, mixed with an equal volume of EtOAc into the separating funnel, and shaken for complete extraction. The secondary compounds contained in the solvent phase were separated from the aqueous phase, dried with Na_2_SO_4_, and evaporated under reduced pressure to yield the crude extracts. The crude extracts were dissolved in 1 mL 2% methanol at a final concentration of 5 mg/mL, the aqueous phase was concentrated 1:10. All fractions were tested against *M. phaseolina* on PDA plates and incubated at 28 ± 2 °C for 5 days.

### 4.7. Stability of Antifungal Metabolites at Different Enzymes, Temperatures and Organic Solvent Conditions

In total, 100 μg/mL of enzymes (trypsin, proteinase K, pancreatin and pepsin) and 10% organic solvents (acetone, ethyl alcohol, chloroform, toluene and isopropyl alcohol) (see Figure 4) were added to 100 µL of culture supernatant. Enzyme-treated samples were incubated for 3 h at 37 °C (42 °C in the case of proteinase K) and the solvent-treated samples were incubated for 3 h at 25 °C and subsequently, 100 µL aliquots were tested for antifungal activity as described above. To assess the stability of the bioactive compounds at high temperatures, CSFs were incubated at 65, 75 and 80 °C for 1 or 3 h, and their activity toward *M. phaseolina* eventually tested.

### 4.8. Size-Fractionated Supernatants Tested for Antifungal Activity

RHFS10 strain was grown in 100 mL of LB broth for 72 h at 28 °C. The cultures were centrifuged at 7000× *g* for 30 min at 4 °C and the supernatants filter-sterilized with a 0.22 µm filter (Millipore, Bedford, MA, USA). The supernatants were size-fractionated (10, 30 and, 50 kDa cutoff spin column; Centricon, Millipore). Fractions were tested for antifungal activity and reported as a percentage of growth inhibition as described above.

### 4.9. LC–MS/MS Analyses

Protein extracts were electrophoretically separated on a 12.5% polyacrylamide gel, under denaturing conditions. Resulting lines were divided into 10 pieces, and each underwent trypsin in gel digestion procedure. NanoUPLC-hrMS/MS analyses of the resulting peptides mixtures were carried out on a Q-Exactive orbitrap mass spectrometer (Thermo Fisher Scientific, Waltham, MA, USA), coupled with a nanoUltimate300 UHPLC system (Thermo Fisher Scientific). Peptides separation was performed on a capillary EASY-Spray C18 column (0.075 × 100, 1.7 μm, Thermo Fisher Scientific) using aqueous 0.1% formic acid (A) and CH_3_CN containing 0.1% formic acid (B) as mobile phases and a linear gradient from 3% to 30% of B in 60 min and a 300 nL min^−1^ flow rate. Mass spectra were acquired over an *m/z* range from 350 to 1500. To achieve protein identification, MS and MS/MS data underwent Mascot software (Matrix Science, London, UK) analysis using the nonredundant Data Bank UniProtKB/Swiss-Prot (Release 2020_03). Parameter sets were: trypsin cleavage; carbamidomethylation of cysteine as a fixed modification and methionine oxidation as a variable modification; a maximum of two missed cleavages; false discovery rate (FDR), calculated by searching the decoy database, ≤0.05. A comparison between the proteins found in the different samples allowed discriminating those specifically expressed by the strains showing promising antifungal activity.

### 4.10. Detection of Antifungal Metabolites

RHFS10 strain was grown in 2 L of LB broth at 28 °C for 72 h with shaking at 150 rpm. The cells were removed by centrifugation (9000× *g*, 30 min) and the supernatant fluid was filter-sterilized using 0.22 μm-pore-diameter membranes. The antifungal activity of the preparation was determined against *M. phaseolina* using the cell-free supernatant assay described above. The culture filtrate (1800 mL) was precipitated with ammonium sulfate (66% *w/v* saturation) and stored overnight at 4 °C with shaking. The precipitate was removed by centrifugation (12,000× *g*, 20 min, 4 °C), resuspended in PBS 1× buffer (0.01 mol/L^−1^, pH 6.5; 1/10 of the initial volume) and dialyzed against the same buffer for 48 h at 4 °C with several changes (dialysis tube, porosity 24, cutoff 12 kDa; Union Carbide Corporation, Danbury, CT, USA). The dialyzed precipitate was lyophilized, and the residue (483 mg) was dissolved in 6 mL ultrapure Milli-Q water and applied to a Sephadex G-50 fine column (Pharmacia, Uppsala, Sweden; 4.5 × 40 cm; flow rate 2.5 mL/min^−1^). The column fractions (3 mL each) were collected in homogeneous groups according to the chromatogram obtained by monitoring proteins concentration at 280 nm [49]. Fractions were lyophilized, tested for antifungal activity (1 mg/dot) against *M. phaseolina*, and analyzed by SDS-PAGE. The SDS-PAGE was performed with 20 μg of total proteins, fractionated on 12.5% SDS polyacrylamide gels and stained by Brilliant Blue Coomassie. Protein concentration was determined with the Bradford assay (Bio-Rad Protein Assay, Hercules, CA, USA; cat no. 500-0006) with bovine serum albumin used as standard.

### 4.11. Minimum Inhibitory Concentrations

The MIC determination was performed in 24-well culture plates according to the method described by Agrillo et al. (2019) [50] with some modification. The wells were prepared in triplicate for each concentration. The retentates (peaks 1, 2, and 3) containing the antifungal compounds were diluted separately at different concentrations (1 mg/mL; 0.5 mg/mL; 200 μg/mL; 100 μg/mL; 50 μg/mL and 25 μg/mL) in a volume of 500 μL of ultrapure Milli-Q water and were inoculated with 500 μL of *M. phaseolina* plugs (4 × 4 mm) resuspended in 2 × PD broth. As a control, 500 μL of *M. phaseolina* plugs (4 × 4 mm) were resuspended in 2 × PD broth diluted with 500 μL of ultrapure Milli-Q water. The retentates were compared with the fungicidal PCNB ≥94% (Sigma-Aldrich) at the same different concentrations. The plates were incubated at 28 °C for 5 days and the MIC was taken as the lowest concentration of antifungal agent at which there was no visible growth of the fungus after incubation. Finally, the percentage of inhibition of the fungal growth was calculated using the formula described above.

### 4.12. Whole-Genome Sequencing

The most promising bacterial strain, RHFS10, which showed outstanding biocontrol performance, was selected for whole-genome sequencing to obtain future relevant genetic information. DNA extraction was performed using the method described above. Genome sequencing was performed by MicrobesNG (Birmingham, UK) with the genomic DNA library prepared using the Nextera XT library prep kit (Illumina) following the manufacturer’s protocol. Libraries were sequenced on the Illumina HiSeq using a 250 bp paired-end protocol. Reads were adapter trimmed using Trimmomatic 0.30 with a sliding window quality cutoff of Q15 [51] and de novo genome assembly was carried out with SPAdes (version 3.7) via MicrobesNG (University of Birmingham, Birmingham, UK).

### 4.13. Statistical Analysis

All the statistical analyses were performed using GraphPad Prism 8 software. Data were expressed as mean ± SEM. Differences among groups were compared by ANOVA or t-test as indicated in figure legends. Differences were considered statistically significant at *p* < 0.05.

## Figures and Tables

**Figure 1 ijms-22-03324-f001:**
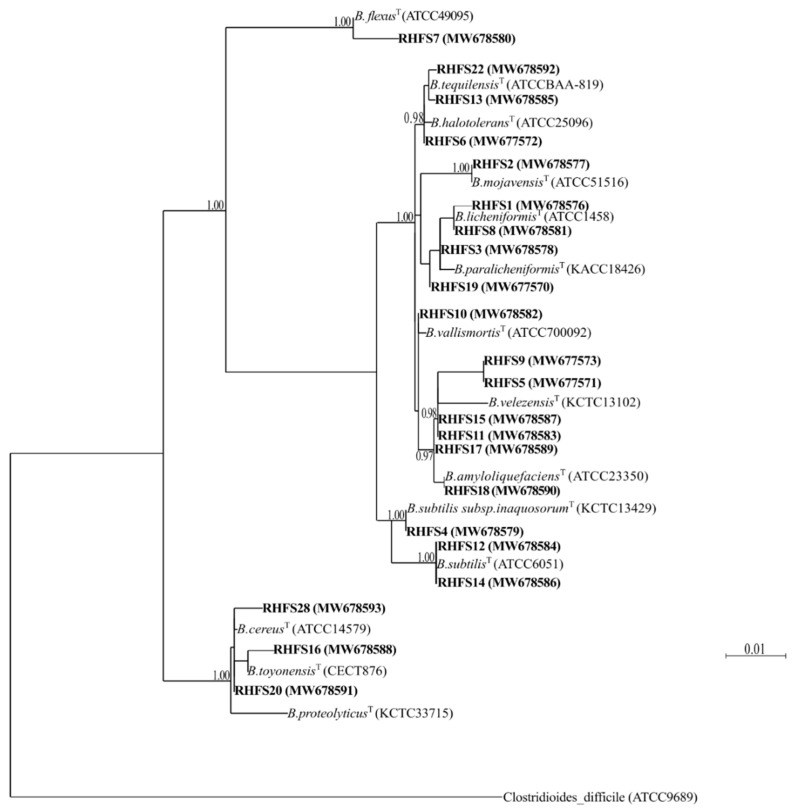
Phylogenetic tree of isolated rhizobacteria. The phylogenetic tree was constructed using the maximum-likelihood algorithm based on 16S rRNA gene sequences. The gene sequences of the isolated bacteria were aligned to the representative type strains (^T^). The numbers in parentheses indicate the GenBank accession numbers. The percentage of replicate trees in which the associated taxa clustered together in the bootstrap test (1000 replicates) is shown next to the branches. The 16S rRNA sequence of *Clostridioides difficile* (ATCC9689) was used to assign an outgroup species.

**Figure 2 ijms-22-03324-f002:**
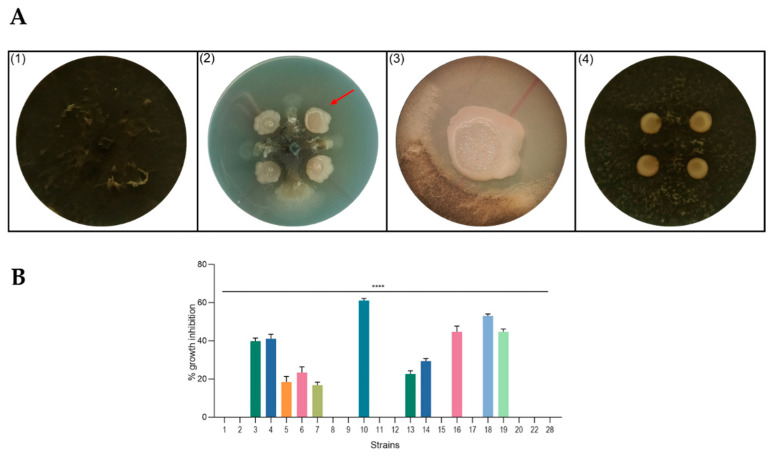
Antagonism assays in solid medium. (**A**) Representative photographs of dual-culture assay for in vitro inhibition of mycelial growth of *M. phaseolina* by isolated strains. (1) *M. phaseolina* (control plate); (2) example of active strain (RHFS10) against *M. phaseolina* growth; (3) images of interaction zone of RHFS10 strain and *M. phaseolina* acquired with a stereoscopic microscope (10× magnification); (4) example of inactive strain (RHFS28) against *M. phaseolina* growth; red arrow in panel 2 indicates the interaction zone magnified in panel 3. (**B**) Inhibition of fungal growth reported as the percentage reduction in the diameter of the fungal mycelia in the treated plate compared to that in the control plate. All experiments were performed in triplicate with three independent trials. Data are presented as means ± standard deviation (*n* = 4) compared to control *M. phaseolina* grown without bacteria. For comparative analysis of groups of data, one-way ANOVA was used and *p* values are presented in the figure: ****: extremely significant < 0.0001.

**Figure 3 ijms-22-03324-f003:**
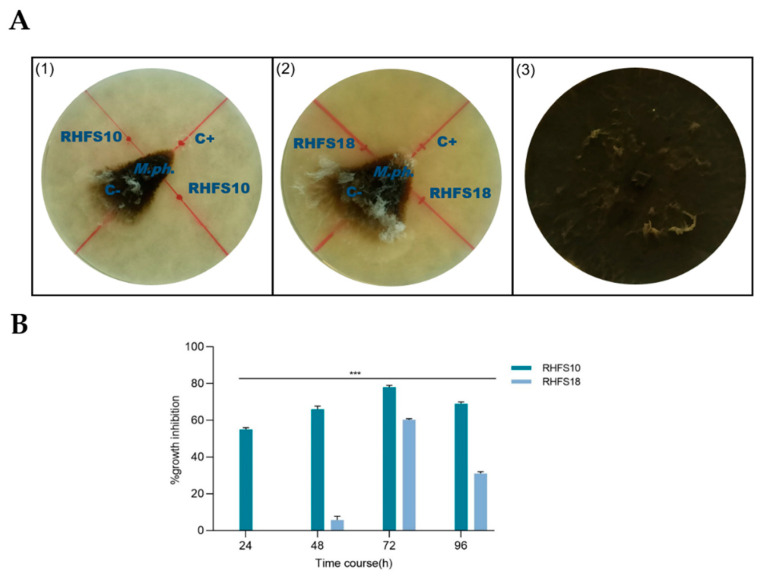
Antifungal activity of secreted metabolites by Plant Growth-Promoting Rhizobacteria (PGPR) strains. (**A**) Effects of the CSFs from RHFS10 (panel 1) and RHFS18 (panel 2) strains collected after 72 h of growth on the mycelial growth of *M. phaseolina* (panel 1). C+: Positive control, pentachloronitrobenzene; C−: Negative control, toluene. All experiments were performed in triplicate with three independent trials. (**B**) Antifungal activity of the Cell-Free Supernatants (CFSs) of the two strains RHFS10 and RHFS18 collected from 24 to 96 h of growth. Percentage of fungal growth inhibition was reported as the percentage reduction in the diameter of the fungal mycelia compared to control plate (panel 3). Data are presented as means ± standard deviation (*n* = 3). For comparative analysis of groups of data, one-way ANOVA was used and *p* values are presented in the figure: ***: extremely significant < 0.001.

**Figure 4 ijms-22-03324-f004:**
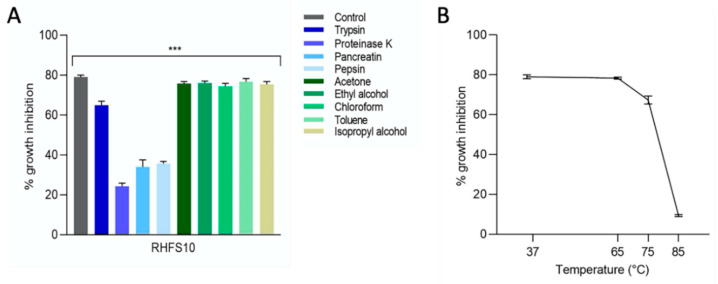
Stability of secreted antifungal metabolites. CFSs collected after 72 h (72-CFS) of RHFS10 was treated separately, with different enzymes and organic solvents (**A**) or incubated at increasing temperatures (37, 65, 75, and 85 °C) (**B**) and tested against *M. phaseolina.* All data represent the average of three separate experiments. ANOVA statistical analysis is extremely significant indicated, *** *p* < 0.001.

**Figure 5 ijms-22-03324-f005:**
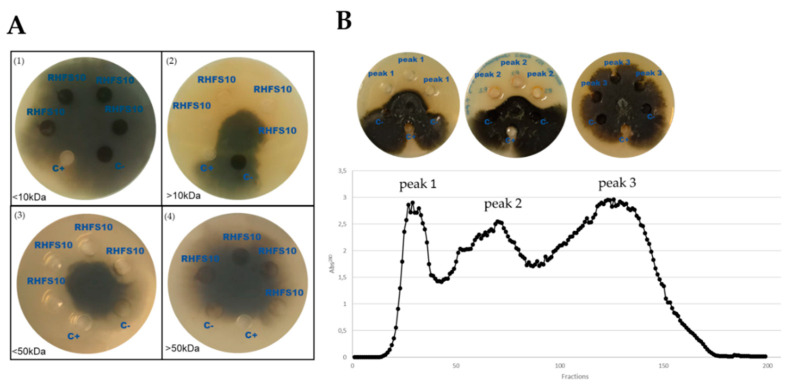
Antifungal activity of cell-free supernatant fractions of RHFS10. (**A**) 72-CFS was size-fractionated using 10, 30 kDa and, 50 kDa cutoff spin columns, and the obtained fractions were tested against *M. phaseolina*. The results obtained with fractions <10 (1), >10 (2), <50 (3) and >50 kDa (4) are reported. C+: Positive control, pentachloronitrobenzene; C−: negative control, toluene; RHFS10: 0.1 mL of fractionated 72-CFS. (**B**) Elution profile of 72-CFS by fractionation on Sephadex G-50 fine column chromatography. The antagonist activity of the three recovered peaks (1 mg/dot) is reported in the upper part of the panel. All data represent the average of three separate experiments.

**Figure 6 ijms-22-03324-f006:**
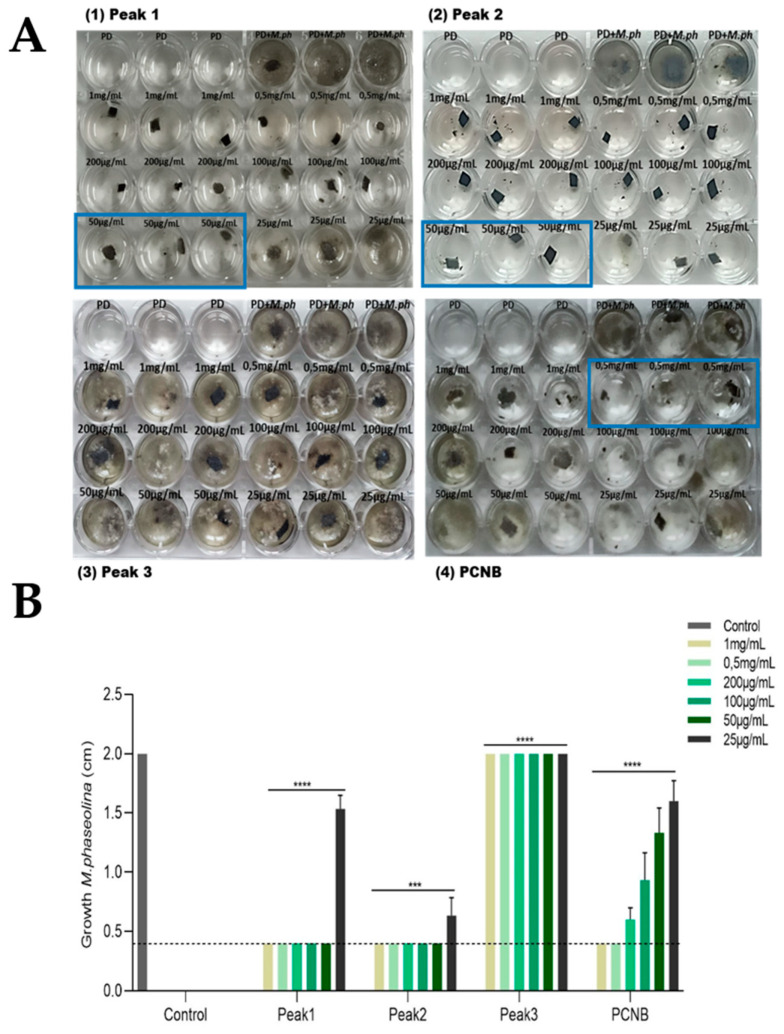
Minimum inhibitory concentrations of purified fractions of 72-CFS on fungal growth. (**A**) Minimum inhibitory concentration of the antifungal compound present in pick 1 (Panel 1), pick 2 (Panel 2) and pick 3 (Panel 3) of purified fractions of 72-CFS using a 24-well plate assay. The commercial fungicide pentachloronitrobenzene (PCNB) (Panel 4) was used as a reference. The tested concentrations are indicated. Fungal plugs incubated with only PD broth (PD + *M. phaseolina*) and the PD alone (PD) were used as a control. The blue lines represent the MICs of the tested samples. (**B**) Graphical representation of the MIC assay. The dotted line indicates the starting size (mm) of *M. phaseolina* plug (4 × 4 mm) at the beginning of the experiment. The results were obtained after 5 days of incubation at 28 °C. Data are presented as means ± standard deviation (*n* = 3 replication for each different concentration). ANOVA statistical analysis is extremely significant indicated—**** *p* < 0.0001 and *** *p* < 0.001.

**Table 1 ijms-22-03324-t001:** Summary of plant growth-promoting and biocontrol traits exhibited by 22 spore-forming bacteria isolates.

Biofertilizer Activities						Biocontrol Activities						
Strains Code	Siderophores Production	PVK *	IAA *	Biofilm	Swarming	Protease Activity	Amylase Activity	Lipase Activity	Xylanase Activity	Cellulase Activity	Chitinase Activity	Catalase Activity
RHFS1	+	-	-	-	-	+++	+++	+	++	++	+	+++
RHFS2	-	++	-	+	+++	+++	+++	+	++	+++	++	-
RHFS3	-	+	+	-	−	+++	+++	++	+	+++	-	-
RHFS4	-	+	+	-	-	+++	+++	-	+++	+++	++	+++
RHFS5	+	-	+	-	-	+++	+++	-	+	+	++	++
RHFS6	-	+	++	-	-	+++	+++	-	+++	++	-	+++
RHFS7	-	-	++	-	-	+	-	+	+++	+++	++	+++
RHFS8	++	+	-	-	-	+++	+++	-	++	+	++	+++
RHFS9	+	-	-	+	+++	+++	+++	++	-	+++	++	++
RHFS10	+++	++	+	++	+++	+++	+++	++	+++	+++	++	+++
RHFS11	+	+	+	-	-	+++	+++	+	+++	-	++	++
RHFS12	-	+	+	+	-	-	+++	-	+++	+++	++	+
RHFS13	-	-	++	-	-	+++	-	-	++	-	++	++
RHFS14	-	++	++	-	-	+	+	-	+++	-	+	+++
RHFS15	+	+	+++	-	-	+++	+++	+	++	+++	++	-
RHFS16	+	+	+	+	-	++	++	-	+++	+++	-	-
RHFS17	+	+	+++	-	-	+++	++	-	+	+	++	+++
RHFS18	+++	++	++	+++	++	+++	+++	++	+++	+++	++	+++
RHFS19	++	++	+	+++	++	+++	+++	++	+++	+++	+	+++
RHFS20	+	-	+	-	-	+	++	-	++	++	++	+++
RHFS22	+	+	+	-	++	+++	-	-	+	++	++	-
RHFS28	-	-	-	-	-	+++	+++	-	++	++	++	++

+++: strong activity (formation halo ≥10 mm); ++: moderate activity (5 mm < halo < 10 mm); +: slight activity (halo < 5 mm); −: no activity; PVK *: Phosphate solubilization activity; IAA *: Indoleacetic acid.

**Table 2 ijms-22-03324-t002:** The proteins identified on the three peaks are listed with their accession (AC) numbers and molecular weights.

Fractions	Mass (Da) ^a^	Swiss Prot AC	Significant Sequences	Score	Description
Peak 1	47.924	XYNC_BACIU	18	1776	Glucuronoxylanase XynC OS = *Bacillus subtilis*
	39.483	SUBN_BACNA	5	1080	Subtilisin NAT OS = *Bacillus subtilis subsp. natto*
	27.42	SUBN_BACNA	5	865	Subtilisin NAT OS = *Bacillus subtilis subsp. natto*
	75.961	SACC_BACSU	1	795	Levanase OS = *Bacillus subtilis*
	38.141	PEL2_BACIU	3	566	Pectin lyase OS = *Bacillus subtilis*
Peak 2	27.365	GUB_BACAM	8	990	Beta-glucanase OS = *Bacillus amyloliquefaciens*
	39.483	SUBN_BACNA	5	800	Subtilisin NAT OS = *Bacillus subtilis subsp. natto*
	27.42	SUBN_BACNA	5	637	Subtilisin NAT OS = *Bacillus subtilis subsp. natto*
Peak 3	72.39	AMY_BACSU	1	41	Alpha-amylase OS = *Bacillus subtilis*
	34.106	MPR_BACSU	1	39	Extracellular metalloprotease OS = *Bacillus subtilis*

^a^ Molecular mass of the Swiss Prot sequence in the absence of molecule processing.

## Data Availability

Not applicable.

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
