# Peer review of "Plant Growth Promotion Function of Bacillus sp. Strains Isolated from Salt-Pan Rhizosphere and Their Biocontrol Potential against Macrophomina phaseolina"

_ijms, 2021, doi:10.3390/ijms22073324_

Round 1

Reviewer 1 Report

The manuscript is an interesting start of the characterization of potential plant growth promotion and biocontrol abilities for finally two Bacillus sp. In the title "Growth promotion ability" shoudl be changed to "Plant growth promotion potential". In the Abstract (line 4), "Recently" is not correct, because since at least two decades, biofertilizer and effective biocontrol strains for fungal pathogens within the Bacillus growth were successfully studied and commercially applied since about at ten years (. Of course, even more research and new isolates are necessary especiialy for specific pathogens to replace or assist industrial plant protection chemicals. 

In the phylogenetic analysis (16S rDNA) of the studied Bacillus isolates, important comparison with the reference / model biocontrol rhizobacterium Bacillus. amyloliquefaciens subsp. plantarum FZB42 (Chowdhury et al., 2015, Front. Plant Science 6, 205), renamed as B. venezensis FZB42  (Fan et al. 2017, Front. Microbiol. 8: 22)  is missing. This strain is already in successful biotechnological applications for many years and is characterized in very much detail and of course completely sequenced and annotated. Therefore a comparison also oft he phenotypic properties is a valid aspect in any new chacterization and description of biocontrol Bacillus strains. For example, is there any IAA-production by the tested newactive isolates?

Specific aspects:

Title, Abstract, Introduction: It should be shortly explained, why  salt-pan plant Juniperus sabina was used for teh new isolation. The name of the plant should appear in the title like „salt-pan rhizosphere of Juniperus sabina…“.

A total of 22 isolates were screened, but only three were studied in detail in the manuscript. Therefore, Table 2 could go into the Supplementum and possibly also other informations concerning the preliminary screening oft he 22 strains.

Figure S2: The CAS-based siderophore plate color essay produces a red to orange color (need to be corrected, since a yellow color is caused only by an acidification around the tested colony).

Author Response

On behalf of all Authors, I would like to thank you and the Reviewers for the positive consideration of the manuscript. All comments and suggestions have been considered and we feel they helped in improving the quality of the manuscript.

specifically:

Point-by-point reply

The manuscript is an interesting start of the characterization of potential plant growth promotion and biocontrol abilities for finally two Bacillus sp.

REPLY: Thank you

R: In the title "Growth promotion ability" shoudl be changed to "Plant growth promotion potential"

REPLY: the title has been changed as suggested

R: In the Abstract (line 4), "Recently" is not correct, 

REPLY: the word "Recently" in the abstract has been replaced.

R: In the phylogenetic analysis (16S rDNA) of the studied Bacillus isolates, important comparison with the reference / model biocontrol rhizobacterium Bacillus. amyloliquefaciens subsp. plantarum FZB42 (Chowdhury et al., 2015, Front. Plant Science 6, 205), renamed as B. venezensis FZB42  (Fan et al. 2017, Front. Microbiol. 8: 22)  is missing. This strain is already in successful biotechnological applications for many years and is characterized in very much detail and of course completely sequenced and annotated. Therefore a comparison also oft he phenotypic properties is a valid aspect in any new chacterization and description of biocontrol Bacillus strains

REPLY: Thank you for useful suggestions. We have modified the phylogenetic three of figure 1 including the B. venezensis FZB42.

R: This strain is already in successful biotechnological applications for many years and is characterized in very much detail and of course completely sequenced and annotated. Therefore a comparison also oft he phenotypic properties is a valid aspect in any new chacterization and description of biocontrol Bacillus strainsFor example, is there any IAA-production by the tested newactive isolates?

REPLY: IAA production has been tested for the 22 new isolates and the B. venezensis FZB42 strain used as model for the biocontrol traits. We also looked for the antifungal lipopetides described in te suggested papers, but it is not produced by our strain.

R: Title, Abstract, Introduction: It should be shortly explained, why  salt-pan plant Juniperus sabina was used for teh new isolation. The name of the plant should appear in the title like „salt-pan rhizosphere of Juniperus sabina…“.

REPLY: the text has been modified and a paragraph about the reason for the choice to select rhizobacteria of J. sabinaadded in the discussion section. We prefer to not further modified the title because afraid that it could be too long.

R: A total of 22 isolates were screened, but only three were studied in detail in the manuscript. Therefore, Table 2 could go into the Supplementum and possibly also other informations concerning the preliminary screening oft he 22 strains.

REPLY: Tables 2 has been modified and moved in Suppl. mat. section together withh table 1 and, table 3 and 4 merged.

R: Figure S2: The CAS-based siderophore plate color essay produces a red to orange color (need to be corrected, since a yellow color is caused only by an acidification around the tested colony)

REPLY: we followed the method described Pérez-Miranda  (2007) to detected siderophores production. As reported in the paper, the color change of the medium depends on the siderophores type produced by bacteria. The color-changing from blue to purple observed for thhr strain RHFS10 and to yellow for the strain RHFS18 of figure S2 are associated with the production of catechol and carboxylate respectively. The strain RHFS28 did not induce any change of color and was used as a negative control

Reviewer 2 Report

The manuscript is interesting and it is quite well written. Unfortunately, there are several flaws and points to be better explained:

Introduction should be extended, enriched. Mainly those parts regarding to Bacillus and M.phaseolina

Bacterial identification: it is mandatory to show the 16S sequences obtained from the bacterial isolates. Table 2 shows the accession numbers of the most similar strains already deposited by other authors. Moreover, the most similar strain has to be referred to a type strain.

Figure 1 needs to be edited. The isolated strains must appear followed by the accession numbers not by the most similar strain. The tree has to be constructed only with the isolated strains and the most similar type strains, type strains need to be marked…etc .

Results should be sorted. Thera are very long tables all along the ms. For example, Table 2 can be deleted (once Figure 1 shows all the data), others can be merged..

Some parts of the methodology are not correct. Authors have used the method described by Khamna et al (2009) but with only one bacteria strike on the plate, so the results are not very clear. Figure 2A panel 2 does not show a 90% of mycelia inhibition. In Figure S4 the best inhibition is showed by the first bacteria of the upper panel but the name is missed. In this figure I cannot see inhibition of mycelia growth by strains RHF1, -5 and -28. Sincerely, I can see a 80% of mycelia inhibition at any figure coming from in vitro assays. These tests should be repeated in the way Khamna et al described or by inoculating four droplets (5μl) of each bacterial culture distributed 1.5 cm from the centre (where the fungal plug has been deposited) in four perpendicular directions. Also for the CSFs assays, the results should be shown with the fungal plug in the centre and four droplets all around. Otherwise the inhibition cant be seen (Figure 3A)

All over the ms the final “S” of the strains’ name is missed (RHF28 instead RHFS28)

Discussion: the first part of sounds quite similar to introduction, please rewrite it

In discussion authors suggest that “the antagonist activity of RHF10 was not influenced by the bacterial growth stage, suggesting a constitutive production of the antimicrobial molecules” while in results they assessed “The antifungal activity increased proportionally with the growth time reaching a maximum after 72 hours”. Please, clarify this question.

Authors affirm strain RHFS10 showed the best performance for plant growth-promoting applications both as bio-fertilizer and biocontrol agent. They should make an effort to go ahead in this characterization and to include plant assays for testing the selected bacteria, at least under greenhouse or growth chamber conditions.

I have pointed additional comments all along the ms.

Author Response

On behalf of all Authors I would like to thank you and the Reviewers for the positive consideration of the manuscript. All comments and suggestions have been considered and we feel they helped in improving the quality of the manuscript.

Specifically:

R:The manuscript is interesting and it is quite well written.

REPLY: Thank you.

R:Unfortunately, there are several flaws and points to be better explained:

Introduction should be extended, enriched. Mainly those parts regarding to Bacillus and M.phaseolina

REPLY: The introduction has been extended as required. Thank you for the useful help.

R: Bacterial identification: it is mandatory to show the 16S sequences obtained from the bacterial isolates. Table 2 shows the accession numbers of the most similar strains already deposited by other authors. Moreover, the most similar strain has to be referred to a type strain.

REPLY: we agree with the reviewer. Table 2 has been completely modified following the Reviewer's suggestions. In particular, the accession numbers of our isolates have been reported instead of ones of the type strains.

R: Figure 1 needs to be edited. The isolated strains must appear followed by the accession numbers not by the most similar strain. The tree has to be constructed only with the isolated strains and the most similar type strains, type strains need to be marked…etc .

REPLY: thank you for the useful suggestion. The figure 1 has been modified as required, in particular, the phylogenetic analysis has been performed only with the isolated strains and the most similar type strains and the accession numbers of the isolates have been reported in the three.

R: Results should be sorted. Thera are very long tables all along the ms. For example, Table 2 can be deleted (once Figure 1 shows all the data), others can be merged..

REPLY: the new table 2 has moved in supplemental material as suggested togrter with table 1. The table 3 and 4 were merged.

R: Some parts of the methodology are not correct. Authors have used the method described by Khamna et al (2009) but with only one bacteria strike on the plate, so the results are not very clear. Figure 2A panel 2 does not show a 90% of mycelia inhibition.

REPLY: the experiments have been repeated following the method described by Khamma et al., 2009.

R: In Figure S4 the best inhibition is showed by the first bacteria of the upper panel but the name is missed.

REPLY:Thank you for the right observation, the correct name of the strain RHFS18 has been added.

R: In this figure I cannot see inhibition of mycelia growth by strains RHF1, -5 and -28.  Strains RHFS1 and RHFS28 have not any anti fungal activity as shown in the grapf of figure 2B. A slight ai

REPLY: Figure S4 supports the graph of figure 2B, reporting some examples of the fungal growth inhibition activity of the isolates. In particular RHFS18, 19, 3, 5, 16 and 4 strains were chosen as able to inhibit the fungal growth, while strains RHFS1 and 28 as not able to affect the fungus.

R: Sincerely, I can see a 80% of mycelia inhibition at any figure coming from in vitro assays. These tests should be repeated in the way Khamna et al described or by inoculating four droplets (5μl) of each bacterial culture distributed 1.5 cm from the centre (where the fungal plug has been deposited) in four perpendicular directions. Also for the CSFs assays, the results should be shown with the fungal plug in the centre and four droplets all around. Otherwise the inhibition cant be seen (Figure 3A)

REPLY: thank you for the useful suggestion. As mentioned upper, the Dual culture assay and CFSs assays have been repeated, and figures 2 and 3 have been changed as required.

R: All over the ms the final “S” of the strains’ name is missed (RHF28 instead RHFS28)

REPLY: DONE

R: Discussion: the first part of sounds quite similar to introduction, please rewrite it

REPLY: DONE

R: In discussion authors suggest that “the antagonist activity of RHF10 was not influenced by the bacterial growth stage, suggesting a constitutive production of the antimicrobial molecules” while in results they assessed “The antifungal activity increased proportionally with the growth time reaching a maximum after 72 hours”. Please, clarify this question.

REPLY: we are in part agree with the reviewer: In the results section we wrote “The antifungal activity increased proportionally with the growth time reaching a maximum after 72 hours,... (Figure 3B) “ referring to both analysing strains RHFS10 and 18, while in the discussion we referred only to strain RHFS10.  A sentence has been added in the text to clarify this point.

In particular in the results section, the sentence has been modified as follows: "The antifungal activity increased proportionally with the growth time reaching a maximum after 72 hours, specifically for the RHFS18 strain (Figure 3B). Based on the efficiency of fungal growth inhibition and on the steadier production of the active secondary metabolites,  strain RHFS10 was chosen for further investigation"

R: Authors affirm strain RHFS10 showed the best performance for plant growth-promoting applications both as bio-fertilizer and biocontrol agent. They should make an effort to go ahead in this characterization and to include plant assays for testing the selected bacteria, at least under greenhouse or growth chamber conditions.

REPLY: We agree with the Reviewer that plant assays could be useful for the manuscript but we believe that the goal of our work is the isolation and characterization of spore-forming PGPRs and represents a first step for the characterization of RHFS10 strain. Moreover, the partial purification and the identification of the active molecules produced by RHFS10 provide new information on antifungal active compounds of interest to the scientific community. Greenhouse experiments and purification of active metabolites have been planned to complete the characterization of  RHFS10.  Therefore, we would prefer to keep the manuscript in its form.

R: I have pointed additional comments all along the ms.

PAGE3 :

R: Table 1: Is this data correct? for the bacteria isolation plates were incubated at 30ºC so that is impossible to grow just at 40ºC

REPLY:The reviewer is right, it was a formatting-mistake and “range 15-40” has been added

Page 5:

R: It is not a PGP feature. Delete it efficiently scavenge H2O2

REPLY:  we agree with the Reviewer. The catalase activity is involved in the biocontrol activity of the PGPR (i.e. "K.Geeth et al., Int.J.Curr.Microbiol.App.Sci (2014) 3(6) 799-809")  and not as a biofertilizing feature. This characteristic has been moved into screening for biocontrol activities..

Page 7:

R: Not described in M&M:

REPLY:  thank you for the correction, the method used to evaluate biosurfactant production was wrongly inserted in paragraph 4.5.1.

R: Strains should be tested for IAA production.

REPLY:  The reviewer is right, the IAA production has been evaluated and the new data added in table2.

R: Biosurfactant production: It is a desirable property for some industries but it is not a PGP trait. 

REPLY:we are in part agree with the Reviewer: biosurfactants production is not considered one of the most fundamental features of the PGPR, nevertheless, several papers refer to Biosurfactant as " effector molecules in plant–microbe interactions, pathogenesis, and phytostimulation which can be beneficial either for the bacteria or for the crops so as to warrant sustainability " (Primo et al., 2015). Moreover, it has been shown that the application of biosurfactant-producing bacteria increases the bioavailability of organic compounds in the soil (Ferajani et al., 2018). So we'd prefer to leave this analysis in our work.

Page 7:

R: I can not understand how the bacteria can inhibit the mycelia growth in the upper zone while it is growing so close to the bacteria in the lower zone of the dish (about figure2A, panel 2)

REPLY: We’ve speculated that bacteria could secrete molecules able to inhibit mycelial growth. To better investigate it, we have repeated the dual culture assay experiment following the suggestions of the Reviewer and the images of figure 2 have been replaced with new ones.

R: Please, add an arrow in panel A2 showing were this magnification came from (about figure2A, panel 4) .... There is not correlantion between a 90% of mycelia growth inhibition and panel A2.

REPLY:  figure 2 has been replaced with a new one and a new graph added.

Page 8:

fig3: this assay should be done each CFS in separate

REPLY: the CSF experiments have been repeated as requested by the reviewer

RHFS28 is missed in the Figure

REPLY: RHFS28 has been not reported in the new graf since the strain has not antifungal activity.

Page 9:

R: Replace "Table 4" by "Figure 4A"

REPLY: DONE

R: lipase is not present in the figure

REPLY: the reviewer is right the sentence has been corrected.

R: Figure 4B

REPLY: DONE

Page 11:

R: Replace by "Figure 6", as assessed in the text

REPLY: DONE

Page 13:

 R: "K" in lower case

REPLY: DONE

R: Although there are some missing results, the antifungal metabolites ranged between 10 and 50kDa

REPLY: the reviewer is right, the value 20 has replaced with the correct one.

Round 2

Reviewer 2 Report

The ms is now acceptable for publication altough some minor points have to be corrected (see pdf). 
